# Correlation between Hypoperfusion Intensity Ratio and Functional Outcome in Large-Vessel Occlusion Acute Ischemic Stroke: Comparison with Multi-Phase CT Angiography

**DOI:** 10.3390/jcm11185274

**Published:** 2022-09-07

**Authors:** Zhifang Wan, Zhihua Meng, Shuangcong Xie, Jin Fang, Li Li, Zhensong Chen, Jinwu Liu, Guihua Jiang

**Affiliations:** 1The Second School of Clinical Medicine, Southern Medical University, Guangzhou 510515, China; 2The Department of Medical Imaging, Guangdong Second Provincial General Hospital, Guangzhou 510317, China; 3Department of Medical Image Center, Yuebei People’s Hospital, Shaoguan 512025, China

**Keywords:** hypoperfusion intensity ratio, CT Perfusion, multi-phase CT angiography, large vessel occlusion, acute ischemic stroke, functional outcome

## Abstract

**Background and purpose:** Previous studies have shown that Hypoperfusion Intensity Ratio (HIR) derived from Perfusion Imaging (PWI) associated with collateral status in large-vessel occlusion (LVO) acute ischemic stroke (AIS) and could predict the rate of collateral flow, speed of infarct growth, and clinical outcome after endovascular treatment (EVT). We hypothesized that HIR derived from CT Perfusion (CTP) imaging could relatively accurately predict the functional outcome in LVO AIS patients receiving different types of treatment. **Methods:** Imaging and clinical data of consecutive patients with LVO AIS were retrospectively reviewed. Multi-phase CT angiography (mCTA) scoring was performed by 2 blinded neuroradiologists. CTP images were processed using an automatic post-processing analysis software. Correlation between the HIR and the functional outcome was calculated using the Spearman correlation. The efficacy of the HIR and the CTA collateral scores for predicting prognosis were compared. The optimal threshold of the HIR for predicting favorable functional outcome was determined using receiver operating characteristic (ROC) curve analysis. **Results:** 235 patients with LVO AIS were included. Patients with favorable functional outcome had lower HIR (0.1 [interquartile range (IQR), 0.1–0.2]) vs. 0.4 (IQR, 0.4–0.5)) and higher mCTA collateral scores (3 [IQR, 3–4] vs. 3 [IQR, 2–3]; *p* < 0.001) along with smaller infarct core volume (2.1 [IQR, 1.0–4.5]) vs. (15.2 [IQR, 5.5–39.3]; *p* < 0.001), larger mismatch ratio (22.9 [IQR, 11.6–45.6]) vs. (5.8 [IQR, 2.6–14]); *p* < 0.001), smaller ischemic volume (59.0 [IQR, 29.7–89.2]) vs. (97.5 [IQR, 68.7–142.2]; *p* < 0.001), and smaller final infarct volume (12.6 [IQR, 7.5–18.4]) vs. (78.9 [IQR, 44.5–165.0]; *p* < 0.001) than those with unfavorable functional outcome. The HIR was significantly positively correlated with the functional outcome [r = 0.852; 95% confidence interval (CI): 0.813–0.884; *p* < 0.0001]. The receiver operating characteristic (ROC) analysis showed that the optimal threshold for predicting a favorable functional outcome was HIR ≤ 0.3 [area under the curve (AUC) 0.968; sensitivity 88.89%; specificity 99.21%], which was higher than the mCTA collateral score [AUC 0.741; sensitivity 82.4%; specificity 48.8%]. **Conclusions:** HIR was associated with the functional outcome of LVO AIS patients, and the correlation coefficient was higher than mCTA collateral score. HIR outperformed mCTA collateral score in predicting functional outcome.

## 1. Introduction

Timely endovascular therapy (EVT) can improve the functional outcome in patients with acute ischemic stroke (AIS) due to large-vessel occlusion (LVO) [1,2]. Rapid and accurate assessment of collateral circulation as well as screening of appropriate patients is critical [3,4]. The functional outcome in patients with AIS is affected by several factors, including collateral circulation status, time of reperfusion therapy, recanalization status, post-treatment complications, and comorbidities [5]. Imaging assessment of collateral circulation status has become one of the most widely used and reliable imaging biomarkers recently [6,7]. Robust pial collateral is associated with smaller infarct core, larger ischemic penumbra, and mismatch ratio [3], which can prolong the treatment time window, delay infarct progression, and improve functional outcome [8]. However, present methods to evaluate collateral circulation and predict functional outcome have constraints in terms of accuracy and clinical practicability.

There are several techniques to evaluate collateral circulation status. Computed tomography angiography (CTA), including single-phase CTA (sCTA) and multi-phase CTA (mCTA), have recently been widely used to noninvasively assess collateral circulation status and several scoring systems such as, Tan, Maas, and Menon have been established [9,10]. Traditional single-phase CTA is simple and rapid but provides only one snapshot, which may lead to underestimation of the collateral status because of the influence of intravascular contrast agent transit time [11]. Multiphase CTA (mCTA), which includes the arterial phase, peak venous phase, and late venous phase, provides time-resolved images and shows the filling range and filling state of collateral vessels, thus, it is a relatively accurate method to evaluate the collateral circulation. The six-point Menon collateral scales [10] has also been widely studied and validated. However, the CTA scoring systems mentioned earlier are easily affected by reader bias and interrater variability and need additional professional training; consequently, CTA cannot provide accurate brain tissue window information [12].

Perfusion imaging provides an estimation of cerebral hemodynamics. The time when the residue function attains its maximum (Tmax) is one of the most commonly used evaluation parameters and has been used to estimate the volume and location of the penumbra in many clinical trials [13]. The hypoperfusion intensity ratio (HIR) is obtained from MR perfusion or CT perfusion and is defined as the quotient between the volumes with Tmax > 10 s and Tmax > 6 s [14]. The areas with Tmax > 6 s include different hypoperfusion degree lesions (ischemic penumbra and severe hypoperfusion) and areas with Tmax > 10 s represent severe hypoperfusion lesions (dangerous tissues lacking collateral blood flow). The proportion of severe hypoperfusion areas will be higher in brain tissue with poor collateral circulation and HIR reflected the collateral status at the level of microperfusion [14,15]. Previous studies have confirmed that HIR was associated with infarction severity, collateral circulation status [16], infarction growth [17], clinical outcome following EVT [14], and patient eligibility before EVT [18] in acute anterior circulation LVO. These studies are majorly based on MR perfusion (partial including CT perfusion). Several recent studies based on CT perfusion have also verified the correlation between HIR and CTA collateral status, proposing that lower HIR is associated with favorable collateral and HIR can be used as a good surrogate to CTA collateral status [19,20,21,22,23]. However, there were some studies which focused on HIR to predict functional outcome in patients with anterior circulation LVO AIS (mainly included patients treated by EVT). Comparative studies of mCTA and HIR to predict the functional outcome were also scarce.

This study’s purpose was to assess whether the HIR derived from CTP could relatively accurately predict the functional outcome in LVO AIS patients received different type of treatment. Additionally, the aim was to obtain the optimal HIR threshold to indicate the good-versus-poor functional outcome and compare the HIR’s ability and the mCTA collateral score to predict the functional outcome.

## 2. Materials and Methods

### 2.1. Study Design and Patient Selection

This retrospective study was authorized by the Ethics Committee of our hospital (SUMC-IRB-2019; 6 March 2019) and the patient need for informed consent was waived owing to the retrospective nature of this study.

Consecutive patients who had acute neurological deficit were confirmed as having AIS with ‘one-stop-shop’ multimodal CT examination within 24 h after onset (time from last known well, TFLKW), were admitted to our single-institution stroke center from January 2019 to July 2021, and were retrospectively reviewed from our prospective database for AIS. The inclusion criteria were as follows: (1) a multimodal CT examination, including unenhanced CT (NCCT), whole-brain volumetric CT perfusion (CTP), and mCTA within 24 h after onset; (2) patients were treated using medical treatment only or with intravenous thrombolysis (IVT), endovascular thrombectomy (EVT), or both; (3) follow-up diffusion-weighted imaging (DWI) or NCCT were performed 24–72 h after standard medical treatment or reperfusion therapy. Exclusion criteria: (1) posterior circulation stroke, (2) without obvious LVO at baseline CTA, (3) poor image quality because of motion artifacts or incomplete acquisition, (4) pre-stroke modified Rankin scale (mRS) score of 2 or greater, (5) without follow-up imaging, and (6) without a 90-d mRS score.

The clinical baseline data of the patients come from the stroke database sustained by the prospective organization of our hospital, which have all medical records, including age, gender, current stroke risk factors, the interval time between onset (time from last known well, TFLKW) to the CT examination, baseline NIHSS (the National Institutes of Health Stroke Scale, NIHSS) score at admission, baseline mRS score before admission, etc. According to the 2018/2019 American Guidelines for Early Management of AIS [24], patients were treated using IVT or/and EVT or stent implantation, while patients with thrombolysis contraindication or unsuitable for EVT were treated using general supportive treatment.

The functional outcome was assessed at 90 d after stroke by a mRS assessor, blinded to the treatment allocation. A favorable functional outcome was defined as an mRS score of 0–2.

### 2.2. Imaging Protocol

Baseline images were obtained using a 128-slice multidetector CT scanner (Discovery CT750 HD; GE Medical Systems, Milwaukee, WI, USA) and included NCCT (section thickness, 5 mm; tube voltage, 120 kV; tube current, 80 mA; field of view (FOV, 250 × 250 mm; matrix 512 × 512), whole-brain volumetric CTP and mCTA. CT perfusion comprised 24 consecutive acquisitions of 55 s (section thickness, 5 mm, pitch 0.992; gantry rotation speed, 0.5 s; tube voltage, 80 kV; tube current, 240 mA; FOV, 250 × 250 mm; matrix, 512 × 512). The mCTA scans were performed eight minutes late after CTP was completed and comprised three phases, with the arch to the vertex coverage in the first (arterial) and skull base to vertex coverage in the second (peak venous) and the third (late venous) phase. Detailed mCTA acquisition parameters were referred to Menon [10] et al.’s study and are as follows: tube voltage 100 kV; tube current automatic mA; rotating speed 0.5 s; section thickness 5 mm, interval 1.25 mm, and overlapping axial images of 1.25 mm. In the first phase, intelligent tracking was used to trigger scanning when the peak of the artery was reached. After the first, the second and the third phase were delayed by 4 s, respectively. During CTP and mCTA scanning, a 50 mL bolus of high-concentration nonionic iodine contrast agent (iopromide, Ultravist 370; Bayer Schering Pharma, Berlin, Germany) was administered at a rate of 5 mL/s and followed by a 30 mL saline flush at the same speed, respectively. MRI was conducted 24–72 h after the onset of symptoms with a 1.5 or 3.0 T-scanner (GE Discovery 750 w; GE Medical Systems). MRI protocol included axial T_1_WI (TR/TE = 1750 ms/24 ms), axial T_2_WI (TR/TE = 5752 ms/93 ms) and T_2_-FLAIR (TR/TE = 6000 ms/160 ms, TI = 250 ms; FOV = 240 mm × 240 mm, layer thickness 5 mm; 1 mm interval between layers) and sequence of axial DWI (6800 ms/90 ms; layer thickness 5 mm; layer spacing 1 mm; NEX 1.0; *b* value = 0 and 1000 sec/mm^2^; FOV = 240 × 240 mm; Matrix128 × 128) and sequence of axial SWI (33.5 ms/37.9 ms; layer thickness 3 mm; layer spacing 1.5 mm; NEX 1.0; flip angle 15°; FOV = 240 mm × 240 mm). For patients with contraindications to MRI or critical illness, follow-up NCCT with the same protocol as baseline CT was conducted instead.

### 2.3. Imaging Reconstruction and Interpretation

All images were evaluated in a consensus reading (by two neuroradiologists).

ASPECTS: The Alberta Stroke Program Early CT Score (ASPECTS) was scored on 5 mm reconstructed axial NCCT images.

mCTA angiograms were reconstructed at a workstation (AW 4.7; GE Healthcare, Milwaukee, WI, USA) with the GE Fast Stroke workflow and multi-plane axial, coronal and sagittal reconstructed images, and axial maximum density projection (MIP) were obtained for all three phases. Axial images with 1 mm overlap and multiplanar axial, coronal and sagittal reconstructions with 3 mm thickness, 1 mm intervals and 1 mm overlap for the first phase were then generated along with axial maximum intensity projections (MIP) for all three phases. The collateral status was graded using the six-point Menon collateral scale (range from 0–5) [10]. A score of 4–5 was defined as favorable collateral and a score of 3–0 was defined as unfavorable collateral or poor collateral. Raters have access to occlusion site, age, and baseline NIHSS, but do not know all other baseline information and CTP results (including HIR). If there was disagreement, the score was completed through negotiation and the consistency test was performed.

The original CTP images were automatically processed using the SHUKUN AI perfusion software (SK-CTPDoc, StrokePro V2.0, SHUKUN Technology, Beijing, China) and six parameter maps of tMIP, CBF (cerebral blood flow), CBV (cerebral blood volume), MTT (mean transit time), TTP (time to peak), and Tmax were obtained rapidly. The infarct core (relative cerebral blood flow, rCBF < 30%) and ischemic area volume (Tmax > 6 s) and mismatch ratio were calculated automatically. Further, the volumes of Tmax > 4 s, Tmax > 6 s, Tmax > 8 s, Tmax > 10 s, and HIR (Tmax > 10 s/Tmax > 6 s) were also calculated automatically. Artery input and venous outflow and threshold was automatically selected using perfusion analysis software. Time–density curve (TDC) was observed and selection of arterial input and venous outflow was manually adjusted when abnormal time density curve was found due to incorrect selection of arteries and veins.

Final infarct volume (FIV) was calculated using a voxel-based method subsuming the entire DWI region (*b* value = 1000 s/mm^2^) hyperintensity or NCCT hypodensity and measured by two resident radiologists with five years of experience and averaged by both. Clinical data were evaluated by a stroke neurologist with six years of experience.

### 2.4. Statistical Analysis

Nominal variables were summarized using frequency descriptive analysis and then compared using the Fisher exact test. Continuous variables were evaluated for normality using histogram analysis and then summarized using mean, standard deviation (SD), median, and interquartile range (IQR) as suitable and then tested using the Mann–Whitney U test. The mCTA collateral score evaluated by each of the two raters were tested for interrater reliability using square-weighted *k* analysis (Cohen *k*). Patients were then dichotomized into two subgroups based on 90-d mRS of favorable functional outcome (mRS ≤ 2) and unfavorable functional outcome (mRS > 2), and the univariate analysis was conducted to compare initial stroke severity assessed using NIHSS score, infarct core volume, Tmax volumes, and perfusion mismatch ratio between subgroups using the Mann–Whitney U test.

The correlations between the HIR and 90-d mRS and the CTA collateral score were calculated using the Spearman correlation. Receiver operating characteristic (ROC) curve analysis was conducted to evaluate the diagnostic performance of the HIR and the CTA collateral score in predicting favorable functional outcome. The Youden index was used to identify the optimal threshold from the ROC curve for predicting favorable functional outcome. The area under the curve (AUC) of each predictor was compared using the Delong test. Univariate and multivariate logistic regression were used to analyze the relationship between HIR and favorable functional outcome. Independent factors for unfavorable functional outcome were evaluated using the multivariate regression model. *p* < 0.05 was considered indicative of a significant difference.

All the analyses were conducted using the statistical software packages R version 3.3.2 (http://www.R-project.org, accessed on 31 October 2016,The R Foundation), Free Statistics software versions 1.4 [25], IBM SPSS version 24 (IBM Corp, Armonk, NY, USA), and MedCalc Statistical Software (version 20; MedCalc Software, Ostend, Belgium).

## 3. Results

### 3.1. Patient Characteristics

A flowchart of patient selection is provided in Figure 1.

In brief, 810 patients with AIS who underwent ‘one-stop-shop’ multimodal CT examination were reviewed from our database. Based on the inclusion/exclusion criteria, patients with posterior circulation stroke (*n* = 236), stroke mimics (*n* = 6), TIA (*n* = 31), moyamoya diseases (*n* = 16), anterior circulation branch-atheromatous diseases (*n* = 214), occlusions of A2/A3 segment or distal M3/M4 segment of anterior cerebral artery (*n* = 25), with poor image quality (*n* = 13), a pre-stroke mRS score of 2 or greater (*n* = 19), absent follow-up imaging (*n* = 10), or absent 90 d mRS score (*n* = 5) were successively excluded. Finally, for analysis, 235 eligible patients (median age, 66 years; interquartile range [IQR], 55.5–73 years; 178 males) were identified.

The patients’ median baseline NIHSS score was 10 [IQR, 5–16]. The median onset (TFLKW) to CT-scanning time was 600 min [IQR, 600–900]. Reperfusion therapy was performed in 118 of 235 patients (50.2%), among whom 31 underwent IVT, 73 underwent EVT, and 14 underwent both. A total of 127 of 235 (54%) patients had favorable functional outcomes. Patients with favorable functional outcomes were younger (median age, 65 years [IQR, 55–72 years] vs. 66 years [IQR, 56–74.2 years]; *p* = 0.54), more likely to be males (81.1% vs. 69.4%; *p* = 0.01), more likely to undergo reperfusion therapy (70/127 (55.1%) vs. 48/108 (44.4%); *p* = 0.34), and had a lower baseline glucose (median, 5.6 [IQR, 6.5–6.8] vs. 6.5 [IQR, 5.7–8.5]; *p* < 0.001) and a lower baseline NIHSS score (median score, 8 [IQR, 4–12] vs. 15 [IQR, 9–19]; *p* < 0.001) than those with unfavorable functional outcomes, and the median onset (TFLKW) to CT-scanning time was lesser (540 [IQR, 330–840]) vs. (660 [IQR, 390–960]; *p* = 0.13).

### 3.2. Association between Baseline Radiologic Features and Functional Outcomes

In the baseline imaging, the median ASPECTS score were 7 [IQR, 6–8]; 85/235 (36.2%) had an occlusion of the internal carotid artery (ICA) or tandem, while 150/235 (63.8%) had an occlusion of the middle cerebral artery (MCA). The median mCTA collateral scores were 3 [IQR,3–4] and the consistency among raters was excellent (*k* = 0.853, 95% CI: 0.796–0.927; *p* = 0.03). The median infarct core volume (rCBF < 30%) was 4.7 mL [IQR, 1.8–17.5] with a Tmax > 6 s perfusion lesion volume of 78.9 mL [IQR, 46.8–121] and mismatch ratio of 13.8 [IQR, 4.6–33.5]. The median HIR was 0.2 [IQR, 0.1–0.4]. FIV at follow-up DWI (210/235, 89.4%) or NCCT was 26.8 mL [IQR, 11.4–76.2]. The median 90 d mRS was 2 [IQR, 1–4].

Patients with favorable functional outcome had lower HIR (0.1 [IQR, 0.1–0.2]; Figure 2) vs. (0.4 [IQR, 0.4–0.5]; *p* < 0.001) and higher mCTA collateral score (median score, 3 [IQR, 3–4] vs. 3 [IQR, 2–3]; *p* < 0.001) along with a smaller infarct core volume (2.1 [IQR, 1.0–4.5]) vs. (15.2 [IQR, 5.5–39.3]; *p* < 0.001) and larger mismatch ratio (22.9 [IQR, 11.6–45.6]) vs. (5.8 [IQR, 2.6–14]); *p* < 0.001), smaller ischemic volume (59.0 [IQR, 29.7–89.2]) vs. (97.5 [IQR, 68.7–142.2]; *p* < 0.001), smaller FIV (12.6 [IQR, 7.5–18.4]) vs. (78.9 [IQR, 44.5–165]; *p* < 0.001) than those with unfavorable functional outcome. Moreover, patients with favorable functional outcomes had more M1/M2 occlusions (82/150 vs. 62/150; *p*
*=* 0.068), while unfavorable functional outcomes had more ICA or tandem occlusion (39/85 vs. 46/85; Figure 3). Patient features, baseline imaging features, clinical and imaging outcomes of the total cohort are summarized in Table 1.

Of the 118 patients who underwent reperfusion therapy, seventy (59%) patients had a favorable outcome. They had lower HIR (0.1 [IQR, 0.1–0.2]) vs. (0.4 [IQR, 0.3–0.4]; *p* < 0.001) and higher mCTA collateral score (median score, 3 [IQR, 3–4] vs. 3 [IQR, 2–3]; *p* = 0.006) along with smaller infarct core volume (2.2 [IQR, 1.1–5.5]) vs. (12.6 [IQR, 5.2–30.4]; *p* < 0.001) and larger mismatch ratio (23.1 [IQR, 11.7–55.8]) vs. (7.0 [IQR, 4.3–19.1]); *p* < 0.001), smaller ischemic volume (70.2 [IQR, 43.4–97.8]) vs. (99.8 [IQR, 72.4–167.8]; *p* < 0.001), smaller FIV (12.1 [IQR, 7.4–17.6]) vs. (87.0 [IQR, 43.4–166.0]; *p* < 0.001; (Appendix A)) than those with unfavorable functional outcome. Of favorable-outcome patients, 45.7% had good collateral (mCTA 4–5); while 53.7% patients who had poor collateral (mCTA 1–3) also obtained favorable outcome due to reperfusion therapy. Of unfavorable-outcome patients, 85% had poor collateral (mCTA 1–3).

Of the 31 patients who underwent IVT, nineteen (61.3%) patients had a favorable outcome. They had lower HIR (0.2 [IQR, 0.1–0.2]) vs. (0.4 [IQR, 0.4–0.5]; *p* < 0.001) and higher mCTA collateral score (median score, 4 [IQR, 3–4] vs. 3 [IQR, 2–3]; *p* = 0.043) along with a smaller infarct core volume (1.6 [IQR, 0.6–3.8]) vs. (11.2 [IQR, 6–28.3]; *p* < 0.001) and larger mismatch ratio (17 [IQR, 13.1–26.9]) vs. (7.6 [IQR, 3–15.9]); *p* = 0.024), smaller ischemic volume (29.2 [IQR, 19.5–72.3]) vs. (70.8 [IQR, 66.1–95.8]; *p* = 0.012), smaller FIV (8.4 [IQR, 5.8–16.1]) vs. (74.7 [IQR, 56.2–142.9]; *p* < 0.001) than those with unfavorable functional outcome (Appendix A). In favorable-outcome patients 63.2% had good collateral (mCTA 4–5); while in unfavorable-outcome patients 83.3% had poor collateral (mCTA 0–3).

Of the 87 patients who received EVT, fifty-one (58.6%) patients had a favorable outcome. They had lower HIR (0.1 [IQR, 0.1–0.2]) vs. (0.4 [IQR, 0.3–0.4]; *p* < 0.001) and higher mCTA collateral score (median score, 4 [IQR, 3–4] vs. 3 [IQR, 3–3]; *p* = 0.04) along with a smaller infarct core volume (2.5 [IQR, 1.2–6.5]) vs. (12.9 [IQR, 5.2–30.4]; *p* < 0.001) and larger mismatch ratio (27.6 [IQR, 10.8–58.9]) vs. (7 [IQR, 4.4–19.1]); *p* < 0.001), smaller ischemic volume (79.4 [IQR, 55.8–114]) vs. (117.9 [IQR, 84–184.7]; *p* = 0.001), smaller FIV (12.8 [IQR, 8–22.4]) vs. (99.2 [IQR, 36.6–169.2]; *p* < 0.001) than those with unfavorable functional outcome (Appendix A). In favorable outcome patients 39.2% had good collateral (mCTA 4–5); while 60.8% patients who had poor collateral (mCTA 0–3) also obtained a favorable outcome due to EVT. While in unfavorable-outcome patients, 77.8% had poor collateral (mCTA 0–3).

Of the 117 patients who only received supportive medical treatment, fifty-seven (48.7%) patients had a favorable outcome. They had lower HIR (0.2 [IQR, 0.1–0.2]) vs. (0.4 [IQR, 0.4–0.5]; *p* < 0.001) and higher mCTA collateral score (median score, 3 [IQR, 3–4] vs. 3 [IQR, 2–3]; *p* < 0.001) along with smaller infarct core volume (1.8 [IQR, 0.9–3.4]) vs. (16.9 [IQR, 6.4–54.7]; *p* < 0.001) and larger mismatch ratio (22.9 [IQR, 12.2–39.8]) vs. (3.8 [IQR, 1.7–11.3]); *p* < 0.001) smaller ischemic volume (42.8 [IQR, 27.4–66.8]) vs. (95.7 [IQR, 66.3–138.9]; *p* < 0.001), smaller FIV (13.8 [IQR, 7.9–20.0]) vs. (77.3 [IQR, 45.2–156.3]; *p* < 0.001; (Appendix A)) than those with unfavorable functional outcome. Of favorable-outcome patients, 52.6% had good collateral (mCTA 4–5); while 85% unfavorable-outcome patients had poor collateral (mCTA 1–3).

### 3.3. Association between HIR, MCTA Collateral Score, and Functional Outcome

There was a significant positive correlation between HIR and functional outcome (Spearman correlation coefficient r = 0.852; 95% CI: 0.813, 0.884; *p* < 0.0001; Figure 4A). The correlation was consistent in various subgroups (Appendix A). There was a moderate negative correlation between mCTA collateral score and functional outcome (r = −0.466; 95% CI: −0.560, −0.359; *p* < 0.0001; Figure 4B) and the correlation coefficient between mCTA collateral score and functional outcome was significantly lower than the correlation coefficient between HIR and functional outcome. There was a moderate negative correlation between HIR and mCTA collateral score (r = −0.518; 95% CI: −0.606, −0.418; *p*< 0.0001; Figure 4C).

At univariate logistic regression analysis, the HIR was significantly associated with an unfavorable outcome not only in total cohort (odds ratio [OR], 1.3 per 0.01; 95% CI, 1.22–1.4; *p* < 0.001; (Table 2) but also in reperfusion treatment arm (OR, 1.35 per 0.01; 95%CI, 1.2–1.52; *p* < 0.001) and supportive medical treatment arm (OR, 1.28 per 0.01; 95% CI, 1.17–1.4; *p* < 0.001). The mCTA collateral score was also significantly associated with functional outcome not only in total cohort (OR, 0.27; 95% CI, 0.18–0.42; *p* < 0.001) but also in reperfusion treatment arm (OR, 0.39; 95% CI, 0.22–0.68; *p* = 0.001) and supportive medical treatment arm (OR, 0.2; 95% CI, 0.1–0.38; *p* < 0.001). Multivariable logistic regression analysis show that the HIR was independently associated with an unfavorable outcome not only in total cohort (adjusted OR [aOR], 1.32 per 0.01; 95% CI, 1.21–1.45; *p* < 0.001) but also in reperfusion treatment arm (aOR, 1.55 per 0.01; 95% CI, 1.21–1.98; *p* < 0.001; (Appendix A)), in the EVT arm (aOR, 1.49 per 0.01; 95% CI, 1.18–1.88; *p* < 0.001; (Appendix A)), and in the supportive medical treatment arm (aOR, 1.31 per 0.01; 95% CI, 1.14–1.5; *p* < 0.001;). The mCTA collateral score was not significantly associated with functional outcome after adjustment both in total cohort (aOR, 0.44; 95% CI, 0.15–1.23; *p* = 0.117) and in reperfusion treatment arm (aOR, 0.2; 95% CI, 0.03–1.48; *p* = 0.117) and in supportive medical treatment arm (aOR, 0.48; 95% CI, 0.11–2.22; *p* = 0.349).

### 3.4. Predictive Ability of HIR and MCTA Collateral Score

ROC analysis revealed that the optimal cut-off point of the HIR to predict an unfavorable outcome was 0.3. The area under the curve (AUC) of the HIR to predict an unfavorable outcome was 0.968 (95% CI: 0.937–0.987; *p* < 0.001; Figure 4D), which was higher than the mCTA collateral score (AUC, 0.741; 95% CI: 0.68–0.795; *p* < 0.001), also the Youden index (0.881 vs. 0.312), sensitivity (88.89% (95% CI: 81.4%–94.1%) vs. 82.4% (95% CI: 73.9%–89.1%)), and specificity (99.21% (95% CI: 95.7%–100%) vs. 48.8% (95% CI: 39.9%–57.8%)) were higher for the HIR than for the mCTA collateral score (Table 3).

## 4. Discussion

Our results indicated a significant positive correlation between HIR and functional outcome and the correlation was consistent in various subgroups. Multivariable logistic regression analysis showed that HIR was independently associated with functional outcome, not only in the total cohort but also in reperfusion treatment arm and supportive medical treatment arm. While the mCTA collateral score was moderately negatively correlated with the functional outcome and the correlation coefficient was significantly lower than the correlation coefficient between HIR and functional outcome. The mCTA collateral score was not independently associated with functional outcome after adjustment. Thus, our research indicated that HIR derived from CT Perfusion (CTP) imaging could relatively accurately predict the functional outcome in LVO AIS patients who received different type of treatment. Additionally, our study identified that the optimal cut-off point of the HIR for favorable outcome was 0.3, and the sensitivity, specificity, and AUC were significantly higher than the mCTA collateral score. Simultaneously, our results also indicated a moderate negative correlation between HIR and mCTA collateral score in patients with acute anterior circulation LVO. Our research primarily focused on the correlation between the HIR and functional outcome in patients with anterior circulation LVO AIS and the efficiency of HIR in predicting the functional outcome, and our study population included patients from the early and late time windows and different type of treatments, which may have essential clinical implications for clinical decision-making and triage of patients for EVT.

Our results showed that a lower HIR (<0.3) was significantly associated with favorable functional outcome, similar to a recent study that suggested that low HIR was related to favorable functional outcome in patients undergoing successful EVT [26]. HIR is the volume of brain tissue with Tmax > 10 s divided by the volume of brain tissue with Tmax > 6 s [14]. Regions of Tmax > 6 s include hypoperfusion lesion of variable degrees. In contrast, regions of Tmax > 10 s stand for severe hypoperfusion regions, and these brain tissue lack collateral flow [14], which may rapidly transform into irreversible infarct core without timely reperfusion treatment [16]. A higher HIR shows a higher proportion of regions of Tmax > 10 s. Typically, a high HIR (>0.4) predicts more rapid growth of the ischemic core, which could limit the therapeutic ability of reperfusion therapies, whereas a large range of Tmax > 10 s regions has been verified to be related to poor outcomes after reperfusion [15]. In our study, the poor functional outcome subgroup contained more patients with HIR > 0.3. These patients showed more brain tissue at risk and poor collateral circulation status, had higher risks of more rapid infarct growth [17,23], infarct volume enlargement, and malignant cerebral edema [21], and were significantly associated with poor functional outcome even received successful EVT. In a recent retrospective study, good HIR (<0.5) was found to be significantly associated with a favorable functional outcome for patients with MCA occlusion achieving TICI 2b recanalization, and the HIR alone may be enough to predict a favorable outcome [27]. In our study, the cut-off point of HIR for a favorable outcome was 0.3, which was lower than the cut-off point (0.5) calculated in the above two recent studies, although direct comparison is limited. This may be because our study included a mixed population of patients with AIS (some only treated with general supportive treatment while others treated with intravenous thrombolysis and/or EVT), whereas the above two recent studies included only patients who underwent successful EVT. Moreover, our research may have a wider scope of application.

The perfusion imaging-based HIR can be used to evaluate collateral circulation status. Our research results also showed that HIR was moderately negatively correlated with mCTA collateral score and a higher HIR indicated a poorer collateral circulation status, which further validated the correlation between HIR and collateral circulation [16,19,22]. A lower HIR indicated a lower proportion of severe hypoperfusion region and rich collateral blood flow in the ischemic region, whereas the favorable collateral blood flow delayed infarct progression and was associated with slower infarct growth, small final infarct volume, and favorable functional outcome [14]. Similarly, a higher HIR showed a higher proportion of severe hypoperfusion brain tissue in the ischemic region; the collateral blood flow was lacking in this region, which indicated poor collateral circulation [14], leading to infarct progression, infarct volume enlargement, and poor functional outcome accordingly. Recent research suggests that a high HIR (>0.5) was associated with poor collateral circulation, infarct progression, and infarct volume enlargement [28], further proving that HIR was related to collateral circulation status. In several recent studies above, the HIR threshold of evaluating favorable collateral circulation was different (0.3–0.5), which might be mainly related to heterogeneities in collateral circulation assessment methods (DSA, sCTA, mCTA) and scoring systems to define favorable collateral circulation [19,20,21,22,23].

According to our research results, the correlation coefficient between HIR and functional outcome was higher than the correlation coefficient between mCTA collateral score and functional outcome. ROC analysis proposed that the sensitivity, specificity, and AUC of the HIR to predicting functional outcome were higher than the mCTA collateral score, indicating that HIR was more accurate and reliable than mCTA collateral score in evaluating the functional outcome. HIR has been shown to be a robust and reliable measure of microvascular collateral flow in stroke, which provides a more nuanced measure of tissue-level collateral blood flow and reflects the brain tissue blood flow conditions in the ischemic region [14]. It is a quantitative parameter with higher accuracy, and accurate collateral assessment enhances the prognosis evaluation performance. Previous studies showed that mCTA collateral score might also be adopted to assess patient prognosis [29], but it was contradictory in some patients, which with favorable mCTA collateral score only had poor functional outcome after received successful EVT [30]. This is because that mCTA only displays the range and filling state of the collateral vessel but does not provide information about the blood volume flowing into the ischemic tissue. Moreover, mCTA collateral score is a qualitative and subjective parameter that is easily affected by the rates [31].

Objective, rapid, and quantitative collateral evaluation is clinically significant in decision-making for patients with acute anterior circulation LVO AIS; as a result, HIR is more advantageous than mCTA in collateral evaluation. First, HIR is a more objective quantitative indicator, eliminating the assessor’s subjectivity. Moreover, it can rapidly acquire related information using automatic analysis software (the post-processing time only need 10–30 s), reducing post-processing time dependence. Additionally, it does not need professional training on the rates, which is particularly applicable to junior doctors and clinical physicians who are unfamiliar with mCTA collateral scoring systems in primary stroke centers. Physicians can rapidly judge the collateral status and make clinical decisions based on HIR. The commercial perfusion post-processing software can rapidly generate a structured electronic report and present the HIR result to the whole multidisciplinary stroke team through the information transfer platform, saving time and accelerating the decision-making process. Therefore, HIR can be applied for determining patient eligibility before EVT [16,18,23].

Notably, on the basis of previous study regarding prognosis prediction model [32] in AIS, the multivariate logistic regression analysis in our study incorporated different variables, including age, gender, blood glucose, baseline NIHSS, baseline ASPECTS, etc. as the influencing factors. HIR was identified as the imaging prediction factor that independently predicted functional outcome. HIR < 0.3 was the optimal threshold that predicted favorable functional outcome and was close to the value of 0.4 confirmed in the previous study [14]. Previous studies showed that low HIR was related to favorable collateral status [19,20,21,22,23], which is consistent with the results of our study.

### Limitations

Because of the limitation of the retrospective nature of our study, our data came from a single center, which inevitably led to selection bias although there was a relatively large sample size. A common CTA scoring method was adopted in our analysis, even though several alternative subjective CTA scoring methods were existing. Further, this study included mainly anterior circulation arteries (extracranial ICA and tandem lesion, M1/M2 lesions), but not other vascular occlusion sites (isolated M3/M4, anterior cerebral artery, posterior cerebral artery, or basilar artery occlusion). The HIR threshold in predicting favorable functional outcomes obtained in this retrospective study should be verified further. It should also be acknowledged that, an AI-based automatic CTP analysis software developed in China, was used, not the well-recognized RAPID software (even though some literature reports that this software also overestimates the ischemic core [33]). Moreover, a specific population was studied, and its generalizability should be further verified. Further, HIR comparison among various perfusion analysis software should also be verified. It is also necessary to further investigate the correlation of HIR with FIV, hemorrhagic transformation, and functional outcome of patients, on which we plan to focus in our next study.

## 5. Conclusions

The HIR was associated with the functional outcome of LVO AIS patients, and the correlation coefficient was higher than mCTA collateral score. HIR outperformed mCTA collateral score in predicting functional outcomes. As a simple, rapid, automatic, and quantitative evaluation approach, HIR is more easily understood than the subjective CTA collateral score and can be employed in clinical decision-making and patient eligibility before EVT.

## Figures and Tables

**Figure 1 jcm-11-05274-f001:**
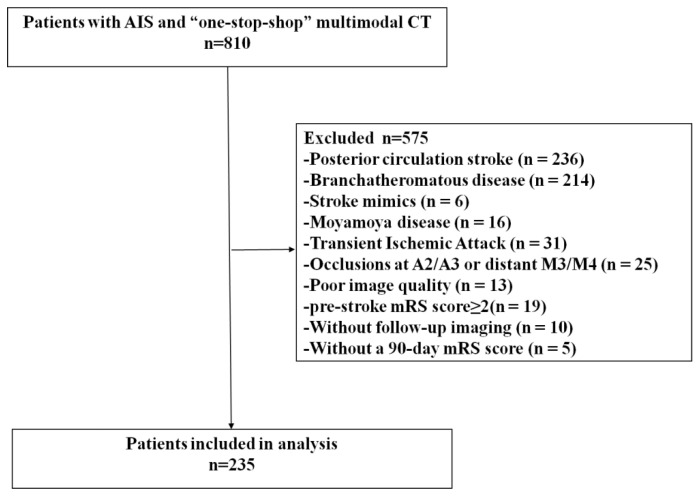
Flowchart of patient selection. AIS, acute ischemic stroke; mRS, modified Rankin scale.

**Figure 2 jcm-11-05274-f002:**
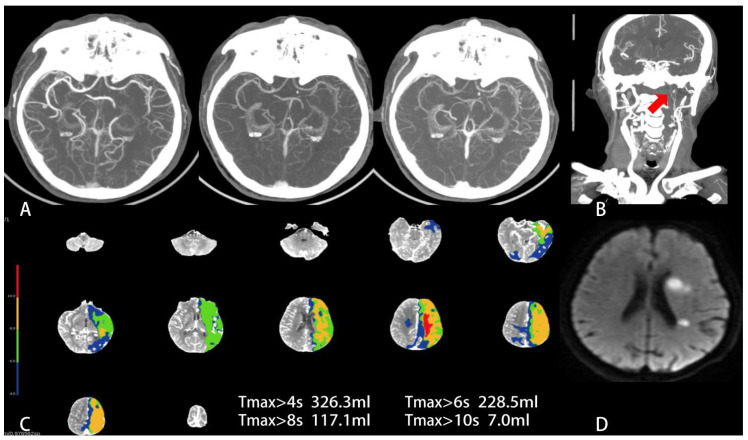
Baseline mCTA, CTP and follow-up DWI in a 70-year-old man with wake-up stroke. The time from last known well to CT scan was 12.5 h. The baseline NIHSS score was 7 and the ASPECTS score on NCCT was 7. (**A**) mCTA axial MIP shows robust pial collateral vessels within the left temporal parietal lobe and only a one-phase delay in the filling of peripheral vessels. The mCTA collateral score was 4 (favorable collateral status). (**B**) Coronal reconstruction of neck CTA MIP shows severe stenosis/occlusion of C1–4 segment of left internal carotid artery (red arrow). (**C**) Tmax map shows the severe hypoperfusion area (red) defined Tmax > 10 s was 7 mL and a hypoperfusion area (light green) defined Tmax > 6 s was 228.5 mL, while HIR (=0.03) was very low. (**D**) This patient underwent a successfully EVT and had a FIV of 8.6 mL on the left basal ganglia and paraventricular region on DWI. The 90 d mRs score of this patient was 2 (favorable functional outcome).

**Figure 3 jcm-11-05274-f003:**
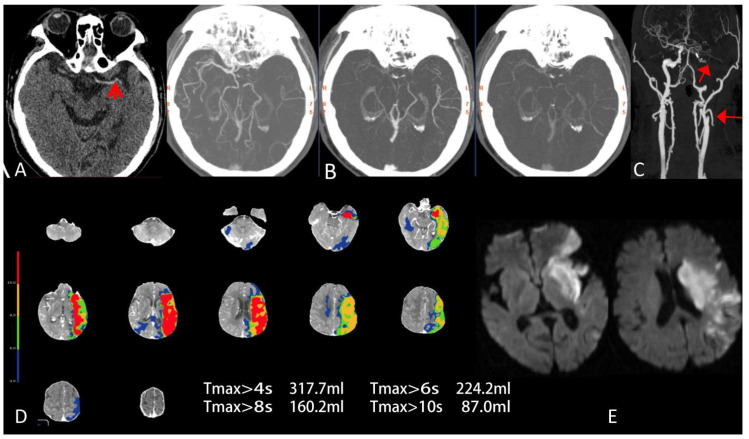
Baseline mCTA, CTP and follow-up DWI in a 79-year-old man with acute ischemic stroke within 6 h after onset and received intravenous alteplase treatment. The time from onset to CT scan was 7.5 h. The baseline NIHSS score was 14 and the ASPECTS on NCCT was 6. (**A**) The high density sign of the middle cerebral artery can be seen in the left temporal lobe on NCCT images (red arrow). (**B**) The mCTA axial MIP shows moderate collateral vessels within the left temporal parietal lobe and a two-phase delay in the filling of peripheral vessels. The mCTA collateral score was 3 (unfavorable collateral status). (**C**) Coronal MIP reconstruction of neck CTA shows severe occlusion of the left internal carotid artery C1 segment to M1 segment of MCA (red arrow). (**D**) Tmax map shows the severe hypoperfusion area (red) defined Tmax > 10 s was relatively large near 87 mL and a hypoperfusion area (light green) defined Tmax > 6 s was 224.2 mL, while HIR (=0.39) was high. (**E**) Family members of the patient refused EVT and this patient had a FIV of 76 mL on the left hemispheric region on DWI. The 90 d mRs score of this patient was 4 (severe disability).

**Figure 4 jcm-11-05274-f004:**
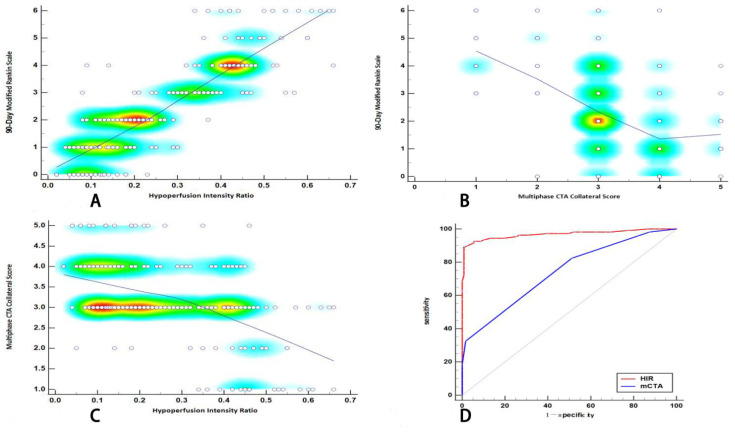
Correlations between HIR, mCTA collateral score and functional outcome, and the receiver operating characteristics. (**A**) Correlation between HIR and functional outcome. (**B**) Correlation between mCTA collateral score and functional outcome. (**C**) Correlation between HIR and mCTA collateral score. (**D**) ROC analysis in the prediction of an unfavorable functional outcome with HIR and mCTA collateral score. Abbreviations: HIR, hypoperfusion intensity ratio; mCTA, multi-phase CT angiography; ROC, Receiver operating characteristic.

**Table 1 jcm-11-05274-t001:** Baseline Patient Characteristics and Comparison of Clinical Variables Stratified by Functional Outcome.

No. of Patients and Characteristics	ALL *n* = 235 (100%)	Favorable Outcome *n* = 127 (54%)	Unfavorable Outcome *n* = 108 (46%)	*p*-Value
Age, Median (IQR)	66 [55.5, 73.0]	65 [55.0, 72.0]	66 [56.0, 74.2]	0.27
Male, *n* (%)	178 (75.7)	103 (81.1)	75 (69.4)	0.054
Risk factors *n* (%)				
Hypertension, *n* (%)	175 (74.5)	90 (70.9)	85 (78.7)	0.221
Diabetes, *n* (%)	61 (26)	26 (20.5)	35 (32.4)	0.054
Hyperlipidemia, *n* (%)	93 (39.6)	52 (40.9)	41 (38)	0.74
Prior stroke, *n* (%)	53 (22.6)	26 (20.5)	27 (25)	0.502
Coronary artery disease, *n* (%)	42 (17.9)	27 (21.3)	15 (13.9)	0.194
Valvular disease, *n* (%)	43 (18.3)	17 (13.4)	26 (24.1)	0.052
Chronic heart failure, *n* (%)	20 (8.5)	7 (5.5)	13 (12)	0.121
Atrial fibrillation, *n* (%)	47 (20)	17 (13.4)	30 (27.8)	0.01
Smoke, *n* (%)	82 (34.9)	50 (39.4)	32 (29.6)	0.154
Drink, *n* (%)	36 (15.3)	22 (17.3)	14 (13)	0.457
Homocysteine, Median [IQR]	13 [10.2, 16.4]	12.9 [10.3, 16.2]	13.2 [10.2, 16.8]	0.457
SBP, Median [IQR]	147 [130, 164.5]	144 [129.5, 165.5]	150 [133, 163]	0.579
DBP, Median [IQR]	87 [78, 96]	87 [78, 96]	88 [79, 98]	0.997
HR, Median [IQR]	79 [72, 89]	78 [71, 86]	80 [75, 96]	0.032
Blood Glucose, Median [IQR]	5.9 [5.2, 7.6]	5.6 [5.0, 6.8]	6.5 [5.7, 8.5]	<0.001
Glycosylated hemoglobin, Median [IQR]	5.8 [5.5, 6.2]	5.8 [5.6, 6.1]	5.8 [5.5, 6.4]	0.515
Time from last known well to CT (min) Median [IQR]	600 [600, 900]	540 [330, 840]	660 [390, 960]	0.134
Baseline NIHSS, Median [IQR]	10 [5, 16]	8 [4, 12]	15 [9, 19]	<0.001
Imaging items				
ASPECTS on NCCT, Median [IQR]	7 [6, 8]	7 [7, 8]	6 [5, 7]	<0.001
Location of occlusion on mCTA, *n* (%)				0.068
ICA or Tandem	85 (36.2)	39 (30.7)	46 (42.6)	
M1	102 (43.4)	56 (44.1)	46 (42.6)	
M2 or further distal	48 (20.4)	32 (25.2)	16 (14.8)	
mCTA score, *n* (%)				<0.001
1	20 (8.5)	0 (0)	20 (18.5)	
2	17 (7.2)	2 (1.6)	15 (13.9)	
3	117 (49.8)	63 (49.6)	54 (50)	
4	63 (26.8)	46 (36.2)	17 (15.7)	
5	18 (7.7)	16 (12.6)	2 (1.9)	
Good collaterals (4–5)	81 (34.5)	62 (26.4)	19 (8)	
Poor collaterals (1–3)	154 (65.5)	65 (27.7)	89 (37.9)	
mCTA score, Median [IQR]	3 [3, 4]	3 [3, 4]	3 [2, 3]	<0.001
ischemic core volume (rCBF < 30%) (mL), Median [IQR]	4.7 [1.8, 17.5]	2.1 [1.0, 4.5]	15.2 [5.5, 39.3]	<0.001
Mismatch ratio, Median [IQR]	13.8 [4.6, 33.5]	22.9 [11.6, 45.6]	5.8 [2.6, 14]	<0.001
TMax > 6 s volume (mL), Median [IQR]	78.9 [46.8, 121]	59.0 [29.7, 89.2]	97.5 [68.7, 142.2]	<0.001
TMax > 10 s volume (mL), Median [IQR]	17.6 [6.3, 39.4]	7.1 [3.1, 13.2]	39.6 [25.3, 65.2]	<0.001
HIR, Median [IQR]	0.2 [0.1, 0.4]	0.1 [0.1, 0.2]	0.4 [0.4, 0.5]	<0.001
FIV, Median [IQR]	26.8 [11.4, 76.2]	12.6 [7.5, 18.4]	78.9 [44.5, 165]	<0.001
Type of treatment, *n* (%)				0.358
Intravenous thrombolysis	31 (13.2)	19 (15)	12 (11.1)	
Bridging therapy	14 (6.0)	7 (5.5)	7 (6.5)	
EVT	73 (31.1)	44 (34.6)	29 (26.9)	
Supportive medical treatment only	117 (49.8)	57 (44.9)	60 (55.6)	
90 d_mRS, Median [IQR]	2 [1, 4]	1 [1, 2]	4 [3, 5]	<0.001

Values in parentheses represent percentage of patients (%); brackets represent first and third quartiles, respectively. Abbreviations: SBP, systolic pressure; DBP, diastolic blood pressure; HR, heart rate; NIHSS, National Institutes of Health Stroke Scale; ASPECTS, Alberta Stroke Program Early Computed Tomography Score; NCCT, unenhanced CT; mCTA, multiphase CTA; ICA, internal cerebral artery; rCBF, relative cerebral blood flow; TMax, the time when the residue function reaches its maximum; HIR, hypoperfusion intensity ratio; FIV, follow-up infarct volume; EVT, endovascular thrombectomy; mRS, modified Rankin scale.

**Table 2 jcm-11-05274-t002:** Multivariable logistic regression analysis for functional outcome.

Variables	Crude OR, 95%CI	*p*-Value	Adjust OR, 95%CI	*p*-Value
Age	1.01 (0.99–1.03)	0.304	1.01 (0.95–1.06)	0.908
Gender	1.89 (1.03–3.46)	0.039	0.94 (0.22–4.05)	0.929
Blood Glucose	1.13 (1.02–1.25)	0.015	1.24 (0.98–1.57)	0.075
NIHSS	1.15 (1.1–1.21)	<0.001	1.04 (0.95–1.15)	0.354
ASPECTS	0.33 (0.25–0.45)	<0.001	0.49 (0.24–1)	0.05
mCTA score	0.27 (0.18–0.42)	<0.001	0.44 (0.15–1.23)	0.117
rCBF < 30%	1.09 (1.05–1.12)	<0.001	0.97 (0.92–1.01)	0.166
TMax > 6 s	1.02 (1.01–1.02)	<0.001	1 (0.99–1.01)	0.955
HIR (per 0.01);	1.3 (1.22–1.4)	<0.001	1.32 (1.21–1.45)	<0.001

Crude model: no other covariates were adjusted. Adjusted model: we adjusted age, gender, blood glucose, NIHSS, ASPECTS, mCTA score, rCBF < 30%, TMax > 6 s. Abbreviations: NIHSS, National Institutes of Health Stroke Scale; ASPECTS, Alberta Stroke Program Early Computed Tomography Score; mCTA, multiphase CTA; rCBF, relative cerebral blood flow; TMax, the time when the residue function reaches its maximum; HIR, hypoperfusion intensity ratio.

**Table 3 jcm-11-05274-t003:** Criterion Values and Coordinates of ROC Analysis Regarding Functional Outcome.

Variable	AUC	95%CI	SE ^#^		Youden Index	Associated Criterion	Sensitivity (%)	95%CI (%)	Specificity (%)	95%CI (%)
HIR	0.968	0.937, 0.987	0.0123	Z = 7.493	0.881	>0.3	88.89	81.4, 94.1	99.21	95.7, 100
mCTA	0.741	0.680, 0.795	0.0288	*p* < 0.0001	0.3123	<3	82.4	73.9, 89.1	48.8	39.9, 57.8

^#^ DeLong et al., 1988. Abbreviations: AUC, area under the curve; CI, confidence interval; ROC, Receiver Operating Characteristic. HIR, hypoperfusion intensity ratio;

## Data Availability

The data presented in this study are available on request from the corresponding author.

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
