# Peer review of "Correlation between Hypoperfusion Intensity Ratio and Functional Outcome in Large-Vessel Occlusion Acute Ischemic Stroke: Comparison with Multi-Phase CT Angiography"

_jcm, 2022, doi:10.3390/jcm11185274_

Round 1

Reviewer 1 Report

 The authors in a single centre study have hypothesized  that HIR derived from CT Perfusion (CTP) imaging could relatively accurately predict the functional outcome in LVO. They have demonstrated that the HIR was more accurate and reliable than mCTA collateral score in predicting the   functional outcome. Thus the  HIR is quick, cost-effective and easy to handle in the clinical setting, thus enabling the clinician to make decisions regarding the end-vascular therapy.The study method, statistical analysis, tables and flow charts are well explained.The limitations of the retrospective nature, the software used to assess the HIR, and external validity of results have been aptly discussed. Further validation is needed in future perspective studies.

Author Response

Reply to Reviewer #1

Dear Reviewer,

Thank you very much for your time involved in reviewing the manuscript and your very encouraging comments on the merits.

Comments:

The authors in a single centre study have hypothesized that HIR derived from CT Perfusion (CTP) imaging could relatively accurately predict the functional outcome in LVO. They have demonstrated that the HIR was more accurate and reliable than mCTA collateral score in predicting the functional outcome. Thus the HIR is quick, cost-effective and easy to handle in the clinical setting, thus enabling the clinician to make decisions regarding the end-vascular therapy. The study method, statistical analysis, tables and flow charts are well explained. The limitations of the retrospective nature, the software used to assess the HIR, and external validity of results have been aptly discussed. Further validation is needed in future perspective studies.

Response:

Thank you very much for your encouraging positive review. Your affirmation is our driving force for further progress. We hope we can be better. Thank you. In our follow-up study, we will design a randomized controlled trial to conduct a prospective cohort study to further verify the reliability of our findings. Sincere thanks for your careful and helpful work.

Reviewer 2 Report

In this study, the authors investigated the correlation between HIR and Functional Outcome in large-vessel Occlusion Acute Ischemic Stroke. They concluded that HIR was associated with the functional outcome of LVO AIS patients, the correlation coefficient was higher than mCTA collateral score and HIR outperformed mCTA collateral score in predicting functional outcome. However, there are still some minor modifications and specific comments listed below.

·          Please check all abbreviations in the text. The author should report in extenso at their first appearance in the text.

·         Please improve the discussion part. Comparisons are needed to justify the results.

·         Can the authors please elaborate on the significance of their findings to patients in future steps?

·         There are language errors that need to be corrected.

Author Response

Dear Reviewer,

Thank you very much for your time involved in reviewing the manuscript and your constructive and detailed comments helpful for improving the quality of this manuscript.

Comments:

In this study, the authors investigated the correlation between HIR and Functional Outcome in large-vessel Occlusion Acute Ischemic Stroke. They concluded that HIR was associated with the functional outcome of LVO AIS patients, the correlation coefficient was higher than mCTA collateral score and HIR outperformed mCTA collateral score in predicting functional outcome. However, there are still some minor modifications and specific comments listed below.

  • Please check all abbreviations in the text. The author should report in extenso at their first appearance in the text.
  • Please improve the discussion part. Comparisons are needed to justify the results.
  • Can the authors please elaborate on the significance of their findings to patients in future steps?
  • There are language errors that need to be corrected.

Comment 1:

Please check all abbreviations in the text. The author should report in extenso at their first appearance in the text.

Response 1:

Thank you for the detailed review and your valuable suggestions to improve the quality of our manuscript. This is a particularly important question that we ignored. Based on your valuable suggestions, we have made some supplements in the corresponding position in the revised manuscript.

Comment 2:

Please improve the discussion part. Comparisons are needed to justify the results.

Response 2:

Thank you for the detailed review and your valuable suggestions to improve the quality of our manuscript. Actually, our study used a ‘one-stop-shop’ multimodal CT examination (including NCCT, mCTA and CTP). Our research primarily focused on the correlation between the HIR and functional outcome in patients with anterior circulation LVO AIS and the efficiency of HIR in predicting the functional outcome, while the advantages and disadvantages of mCTA vs CTP were not the focus of our discussion. We have accepted your valuable suggestions and added relevant contents in the revised manuscript. 

Comment 3:

Can the authors please elaborate on the significance of their findings to patients in future steps?

Response 3:

Thank you once again for your valuable suggestion to improve the quality of our manuscript. We believe that this is an excellent suggestion. Our findings have changed the work flow in our comprehensive stroke center. For patients with acute ischemic stroke more than 6 hours after onset, we only performed NCCT and CT perfusion without additional multi-phase CTA examination to accurately evaluate the state of collateral circulation, and suitable patients (HIR<0.3) can be selected for endovascular treatment, which is helpful to improve the functional outcome of patients. At the same time, multi-phase CTA images can also be obtained by using CTP source images.

Comment 4:

There are language errors that need to be corrected.

Response 4:

Thanks again for your great suggestion on improving the accessibility and quality of our manuscript. We feel so sorry for our poor expressions and writings because English is not our native language. We had done our best to recheck and correct spelling and grammatical errors in the revised manuscript. And we hope the revised manuscript could be acceptable for you. Sincere thanks for your careful and helpful work.

Reviewer 3 Report

This is a very interesting paper exploring correlations between a CT-derived PWI parameter and outcome of patients with LVO AIS. The study is very interesting and clearly exposed. Moreover the results are very interesting for the scientific community. The intrinsic limits of the study are recognized.

I just suggest the authors to add a sentence relative to the radiation dose to the patients (which is obviously increased compared to mCTA alone studies) and another sentence relative to the need of a commercial software, which can limit the widespread use of the score.

Author Response

Dear Reviewer,

Thank you very much for your time involved in reviewing the manuscript and and your very encouraging comments on the merits.

Comments:

This is a very interesting paper exploring correlations between a CT-derived PWI parameter and outcome of patients with LVO AIS. The study is very interesting and clearly exposed. Moreover the results are very interesting for the scientific community. The intrinsic limits of the study are recognized.

I just suggest the authors to add a sentence relative to the radiation dose to the patients (which is obviously increased compared to mCTA alone studies) and another sentence relative to the need of a commercial software, which can limit the widespread use of the score.

Response:

Thank you very much for your encouraging positive review. Your affirmation is our driving force for further progress. We hope we can be better. Thank you.

We think this is an excellent suggestion. Although CTP was obviously increased the radiation dose of the patients compared to mCTA alone studies, the valuable brain tissue window information provided by CTP was indispensable and the increased radiation damage to the patients was negligible. The CTP and multi-phase CTA examination had their unique advantages and disadvantages in clinical diagnosis and treatment decision-making of patients with acute ischemic stroke, and they cannot completely replace each other. The commercial automatic-AI perfusion software we use was currently free, and its applicability needs to be further verified in our future clinical randomized controlled trials. Sincere thanks again for your careful and helpful work.